# Lessons Learned from a Unifying Empirical Study of Parameter-Efficient Transfer Learning (PETL) in Visual Recognition

## Abstract

Parameter-efficient transfer learning (PETL) has attracted significant attention lately, due to the increasing size of pre-trained models and the need to fine-tune them for superior downstream performance. This community-wide enthusiasm has sparked a plethora of approaches. Nevertheless, a systematic study to understand their performance and suitable application scenarios is lacking, leaving questions like *"when to apply PETL"* and *"which approach to use"* largely unanswered, especially in visual recognition. In this paper, we conduct a unifying empirical study of representative PETL approaches in the context of Vision Transformers (ViT). We systematically tune their hyper-parameters to fairly compare their accuracy on downstream tasks. Our study not only offers a valuable user guide but also unveils several new insights. First, if tuned carefully, different PETL approaches can obtain quite similar accuracy in the low-shot benchmark VTAB-1K. This includes simple approaches like fine-tuning the bias terms that were reported inferior. Second, though with similar accuracy, we find that PETL approaches make different mistakes and high-confidence predictions, likely due to their different inductive biases. Such an inconsistency (or complementariness) opens up the opportunity for ensemble methods, and we make preliminary attempts at this. Third, going beyond the commonly used low-shot tasks, we find that PETL is also useful in many-shot regimes — it achieves comparable and sometimes better accuracy than full fine-tuning, using much fewer learnable parameters. Last but not least, we investigate PETL's ability to preserve a pre-trained model's robustness to distribution shifts (*e.g.*, a CLIP backbone). Perhaps not surprisingly, PETL approaches outperform full fine-tuning alone. However, with weight-space ensembles, the fully fine-tuned model can better balance target (*i.e.*, downstream) distribution and distribution shift performance, suggesting a future research direction for PETL.

## 1 Introduction

Pre-training and then fine-tuning has become the standard practice to tackle visual recognition problems (Bommasani et al., 2021). The community-wide enthusiasm for open-sourcing has made it possible to access large, powerful pre-trained models learned from a gigantic amount of data, e.g., ImageNet-21K (Ridnik et al., 2021) or LAION-5B (Schuhmann et al., 2022). More research focus has thus been on how to fine-tune such large models (Yu et al., 2023a). Among existing efforts, parameter-efficient transfer learning (PETL), a.k.a parameter-efficient fine-tuning (PEFT), has attracted increasing attention lately (Han et al., 2024; Ding et al., 2023). Instead of fine-tuning the whole model (*i.e.*, full fine-tuning) or the last fully connected layer (*i.e.*, linear probing), PETL approaches seek to update or insert a relatively small number of parameters to the pre-trained model (Xin et al., 2024). Doing so has several noticeable advantages. First, as named, PETL is parameter-efficient. For one downstream task (*e.g.*, recognizing bird species or car brands), it only needs to learn and store a tiny fraction of parameters on top of the pre-trained model. Second, accuracy-wise, PETL has been shown to consistently outperform linear probing and often beat full fine-tuning, as reported on the commonly used low-shot image classification benchmark VTAB-1K (Zhai et al., 2019).

To date, a plethora of PETL approaches have been proposed, bringing in inspiring ideas and promising results. Along with this come several excellent surveys that summarize existing PETL approaches

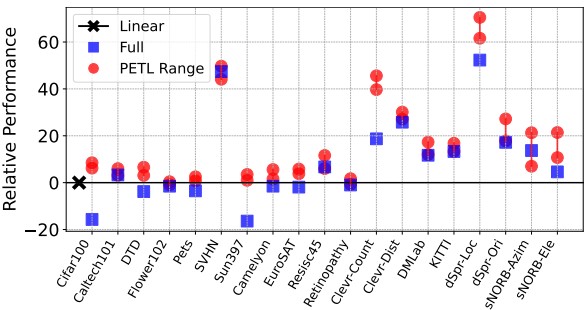 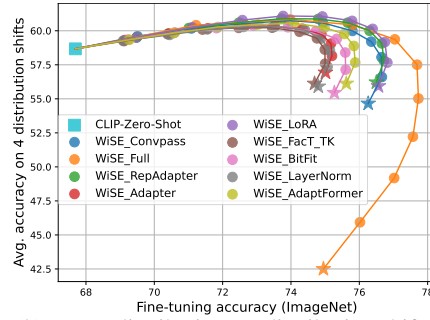

(a) Accuracy gain vs. linear probing on VTAB-1K (19 tasks)   (b) Target distribution vs. distribution shifts

Figure 1: Highlights of our insights. **(a) Downstream accuracy:** if tuned carefully, different PETL methods achieve similar accuracy (●-● for the range from the most to the least accurate methods) and consistently outperform linear probing (×) and full fine-tuning (■) on VTAB-1K. **(b) Distribution shift accuracy:** fine-tuning a CLIP ViT-B/16, known for its generalizability across domains, with PETL on ImageNet-1K (100 samples/class) better preserves the distribution shift accuracy (Y-axis, averaged across ImageNet-V2, ImageNet-S, ImageNet-R, and ImageNet-A) than full fine-tuning, evidenced by the ⋆ points. Interestingly, *weight-space ensembles (WiSE) (Wortsman et al., 2022) is applicable between PETL's fine-tuned model and the pre-trained model (■), but not as effective as applying it to the fully fine-tuned model.* Details are in section 3 and section 7.

(Yu et al., 2023a; Xin et al., 2024; Ding et al., 2023). *Yet, a systematic understanding of the PETL paradigm seems still missing.*

For example, with so many PETL approaches, there is a lack of unifying references for when and how to apply them. Though superior accuracy was reported on the low-shot benchmark VTAB-1K, there is not much discussion on how PETL approaches achieve it. Does it result from PETL's ability to promote transferability or prevent over-fitting? The current evaluation also raises the question of whether PETL is useful beyond a low-shot scenario. Last but not least, besides superior accuracy, do existing PETL approaches offer different, ideally, complementary information?

Attempting to answer these questions, we conduct a unifying empirical study of representative PETL approaches in the context of Vision Transformers (ViT) (Dosovitskiy et al., 2020). This include Low-Rank Adaptation (LoRA) (Hu et al., 2021), Visual Prompt Tuning (VPT) (Jia et al., 2022), Adapter (Houlsby et al., 2019), and ten other approaches. We systematically tune their hyper-parameters to fairly compare their accuracy on the low-shot benchmark VTAB-1K. This includes learning rate, weight decay, and approach-specific parameters like the size of PETL parameters. Besides VTAB-1K, we examine PETL approaches on full-size downstream datasets such as CIFAR-100 (Krizhevsky et al., 2009), RESISC for remote sensing image scene classification (Cheng et al., 2017), and Clevr-Distance for depth classification with synthetic data (Zhai et al., 2019; Johnson et al., 2017). We also conduct a study on ImageNet (Deng et al., 2009) and its variants with domain shifts (Hendrycks et al., 2021a; Gao et al., 2023; Hendrycks et al., 2021b; Recht et al., 2019).

We summarize our key findings and extended analyses as follows.

**Representative PETL approaches perform similarly on VTAB-1K, if properly implemented.** This includes methods previously considered less effective, such as fine-tuning the bias terms (Zaken et al., 2022) in the pre-trained backbone and methods originally proposed for NLP, like Adapter (Houlsby et al., 2019). Among all the hyper-parameters, we find the drop path rate (Huang et al., 2016) quite important. Ignoring it (*i.e.*, setting it to 0) significantly degrades the performance. Overall, PETL approaches consistently outperform linear probing and full fine-tuning on all 19 image classification tasks (each with 1, 000 training examples) in VTAB-1K.

**While similarly accurate on average, PETL approaches make different predictions.** The above finding seems daunting: *if existing PETL approaches all perform similarly in terms of accuracy, do we learn anything useful beyond a single approach?* This is particularly worrisome given that they fine-tune the same backbone using the same downstream data. Fortunately, our analysis shows that different PETL methods learn differently from the same data, resulting in diverse prediction errors and confidence. We attribute this to their difference in inductive biases (Neyshabur et al., 2014) — they explicitly specify different parameters to be updated or inserted. This opens up the door to leverage their discrepancy for improvement, *e.g.*, through ensemble methods (Dietterich, 2000; Zhou, 2012) or co-training (Blum & Mitchell, 1998; Balcan et al., 2004) and we provide preliminary studies.

**PETL is also effective in many-shot regimes.** We apply PETL beyond the low-shot regime and find it effective even with ample downstream training data — PETL can be on par or outperform full fine-tuning. This suggests that varying a fraction of parameters of a properly chosen pre-trained backbone (*e.g.*, pre-trained on ImageNet-21K (Dosovitskiy et al., 2020)) could already offer a sufficient effective capacity (Zhang et al., 2021) to reach a performant hypothesis for downstream tasks.

**PETL is more robust than full fine-tuning to distribution shifts, but with weight-space ensembles, the observation is overturned.** We also evaluate PETL's robustness to distribution shifts, inspired by (Wortsman et al., 2022). We consider a CLIP backbone (Radford et al., 2021), known for its superior generalizability to distribution shifts, and apply PETL to fine-tune it with ImageNet-1K. We find that PETL preserves CLIP's generalizability (*e.g.*, to samples from ImageNet-Sketch or ImageNet-Rendition) better than full fine-tuning. This may not be surprising. What is interesting is that the weight-space ensembles (WiSE) between the fine-tuned and pre-trained models (Wortsman et al., 2022) apply to PETL as well to further improve the robustness without sacrificing the downstream accuracy. Nevertheless, full fine-tuning with WiSE can achieve even higher accuracy in both downstream and distribution shift data than PETL, suggesting a further research direction in PETL.

**What lead to PETL's success?** We attempt to answer this fundamental question by analyzing the results of our study. On VTAB-1K with 19 tasks, we find two cases: 1) in some tasks, full fine-tuning outperforms linear probing, suggesting the need to update the backbone; 2) in some tasks, linear probing outperforms full fine-tuning, suggesting either the backbone is good enough or updating it risks over-fitting. The superior accuracy of PETL in both cases suggests that PETL acts as an *effective regularizer* during low-shot training. Still using VTAB but with ample training data, we find that for tasks in case 1), PETL is on par with full fine-tuning, suggesting that its regularization role does not prevent the fine-tuned model from learning sufficiently from the data. For tasks in case 2), PETL can surprisingly still outperform full fine-tuning, suggesting that it effectively transfers (or preserves) some useful pre-trained knowledge that full fine-tuning may wash away. In sum, PETL succeeds as a **high-capacity learner** equipped with an **effective regularizer**.

**Contributions.** Instead of chasing the leaderboard, we systematically understand existing approaches via a unifying study. Our contribution is thus not a technical novelty, but: (1) a **systematic framework** enabling consistent and reproducible evaluations of PETL methods; (2) a set of **empirical recommendations** on when and how to use PETL methods for practitioners; (3) **new insights for future research** including leveraging PETL's prediction differences and exploring robust fine-tuning.

**What do we not investigate?** There are many aspects that one can ask about PETL. Our study does not consider computation-specific properties like memory usage and FLOPS.

## 2 BACKGROUND

### 2.1 LARGE PRE-TRAINED MODELS

Pre-trained models have become an indispensable part of modern AI development (Bommasani et al., 2021). Building upon neural networks with millions if not billions of parameters and gigantic amounts of training data, these large pre-trained models have led to groundbreaking results in various downstream tasks (Liang et al., 2024; Moor et al., 2023) and shown several emerging capabilities not observed previously (Khan et al., 2022; Li et al., 2024; Bommasani et al., 2021). For example, in computer vision, a Vision Transformer (ViT) (Dosovitskiy et al., 2020) trained with ImageNet-21K (around $14$M images) leads to consistent gains v.s. a ViT trained with ImageNet-1K (around $1.3$M images) (Dosovitskiy et al., 2020). ViTs pre-trained with millions of image-text pairs via a contrastive objective function (*e.g.*, a CLIP-ViT model) (Radford et al., 2021; Cherti et al., 2023) show an unprecedented zero-shot capability and robustness to distribution shifts (Radford et al., 2021). In this paper, we focus on the ImageNet-21K-ViT and use the CLIP-ViT in a robustness study.

**Vision Transformer (ViT).**

We briefly review ViTs (Dosovitskiy et al., 2020), which are adapted from the Transformer-based models (Vaswani et al., 2017) in NLP. ViTs divide an image into a sequence of $N$ fixed-sized patches and treat them like NLP tokens. Each patch is first embedded into a $D$-dimensional vector $x_0^{(n)}$ with positional encoding. The sequence of vectors is then prepended with a "CLS" vector

$\boldsymbol{x}_0^{(\text{Class})}$ to generate the input $\boldsymbol{Z}_0 = [\boldsymbol{x}_0^{(\text{Class})}, \boldsymbol{x}_0^{(1)}, \cdots, \boldsymbol{x}_0^{(N)}] \in \mathbb{R}^{D \times (1+N)}$ to a ViT, composed of $M$ Transformer layers. We use super-/sub-script to index token/layer. The output of the "CLS" token $\boldsymbol{x}_M^{(\text{Class})}$ is used as the image representation.

Each of the ViT's $M$ Transformer layers consists of a multi-head self-attention (MSA) block, a multi-level perceptron (MLP) block, two Layer Normalization (LN) blocks (Ba et al., 2016), and two residual links. The $m$-th Transformer layer can be formulated as

$$\boldsymbol{Z}'_m = \text{MSA}\left(\text{LN}\left(\boldsymbol{Z}_{m-1}\right)\right) + \boldsymbol{Z}_{m-1}, \tag{1}$$

$$\boldsymbol{Z}_m = \text{MLP}\left(\text{LN}\left(\boldsymbol{Z}'_m\right)\right) + \boldsymbol{Z}'_m, \tag{2}$$

where $\boldsymbol{Z}_{m-1} = [\boldsymbol{x}_{m-1}^{(\text{Class})}, \boldsymbol{x}_{m-1}^{(1)}, \cdots, \boldsymbol{x}_{m-1}^{(N)}] \in \mathbb{R}^{D \times (1+N)}$ is the output of the preceding $(m-1)$-th Transformer layer. The MLP is applied to each column vector of $\boldsymbol{Z}'_m$ independently.

Without loss of generality, let us consider an MSA block with a single head. Given a generic input $\boldsymbol{Z} \in \mathbb{R}^{D \times (1+N)}$, this block first projects it into three matrices, Query $\boldsymbol{Q}$, Key $\boldsymbol{K}$, and Value $\boldsymbol{V}$

$$\boldsymbol{Q} = \boldsymbol{W}_Q \boldsymbol{Z}, \quad \boldsymbol{K} = \boldsymbol{W}_K \boldsymbol{Z}, \quad \boldsymbol{V} = \boldsymbol{W}_V \boldsymbol{Z}, \tag{3}$$

where $\boldsymbol{W}_{Q/K/V} \in \mathbb{R}^{D \times D}$ are projection matrices. The output of this block is then formulated as

$$\boldsymbol{V} \times \text{Softmax}(\frac{\boldsymbol{K}^\top \boldsymbol{Q}}{\sqrt{D}}) \quad \in \mathbb{R}^{D \times (1+N)}. \tag{4}$$

### 2.2 PARAMETER EFFICIENT TRANSFER LEARNING (PETL)

Fine-tuning is arguably the most common way to tailor a pre-trained model for downstream tasks. As the size of pre-trained models gets larger, copying and updating all the parameters for one downstream task becomes inefficient. PETL has thus emerged as a promising paradigm.

PETL was originally developed in NLP (He et al., 2021a; Lester et al., 2021; He et al., 2022b; Mao et al., 2022; Sung et al., 2021; Zaken et al., 2022; Asai et al., 2022; Vu et al., 2022; Liu et al., 2022a; Su et al., 2022; Zhong et al., 2022) and has attracted increasing attention in vision (Jia et al., 2022; Chen et al., 2022b; Jie & Deng, 2022; Zhang et al., 2022; Liu et al., 2022b; Lian et al., 2022). Existing approaches can generally be categorized into four groups: prompt-based, adapter-based, direct selective parameter tuning, and efficient selective parameter tuning. *We focus on visual recognition and compare representative PETL approaches applicable to ViTs.* During fine-tuning, all approaches learn a new FC layer for prediction.

**Prompt-based approaches.** Prompt-based learning emerged in NLP (Liu et al., 2023; Lialin et al., 2023). The core concept is to augment the input data with task-specific hints (prompts). **Visual Prompt Tuning (VPT)** (Jia et al., 2022) adapts such an idea to ViTs. Specifically, its deep version (VPT-Deep) prepends a set of soft prompts to the input tokens of each Transformer layer (*i.e.*, $\{\boldsymbol{Z}_m\}_{m=0}^{M-1}$) and only optimizes the prompts during fine-tuning. Other representative works in this category include (Yu et al., 2023b; Tu et al., 2023; Gu et al., 2023).

**Adapter-based approaches.** This category typically introduces additional trainable parameters (*e.g.*, an MLP block) to the frozen pre-trained model (Lialin et al., 2023). It was initially developed for multi-domain adaptation (Rebuffi et al., 2017; 2018) and continual learning (Rosenfeld & Tsotsos, 2018; Mai et al., 2022), and was subsequently extended to the NLP and vision domains to adapt Transformer-based models (Houlsby et al., 2019; Yu et al., 2023b).

We consider five popular adapter-based methods. **Houl. Adapter** (Houlsby et al., 2019) is the first adapter-based PETL approach. It inserts two Adapters — a two-layer bottleneck-structured MLP with a residual link — into each Transformer layer, one after the MSA block and the other after the MLP block. **Pfeif. Adapter** (Pfeiffer et al., 2021) inserts the Adapter solely after the MLP block, a strategy shown effective in recent studies (Hu et al., 2021). **AdaptFormer** (Chen et al., 2022b) inserts the Adapter in parallel with the original MLP block in a Transformer layer, different from the sequential design of Houl. and Pfeif. Adapter. One can view it as an ensemble, summing the task-specific features (by the Adapter) and the task-agnostic features (by the original MLP) to form $\boldsymbol{Z}_m$ in Equation 2. **ConvPass** (Jie & Deng, 2022) introduces a convolutional-based bottleneck module (without a skip link) that explicitly encodes visual inductive biases: the 2D convolution is performed

over tokens of nearby patches. The module is inserted in parallel with the MSA and/or MLP block. **RepAdapter** (Luo et al., 2023) introduces a linear Adapter with group-wise transformations (Luo et al., 2022) and sequentially inserts two such modules after both MSA and MLP blocks.

**Direct selective parameter tuning.** This category selectively updates a subset of parameters of the pre-trained model, seen as a trade-off between full fine-tuning and linear probing. We consider three approaches. **BitFit** (Zaken et al., 2022) updates the bias terms, including those in the Q/K/V projections, the MLP blocks, the LN blocks, and the projection for patch embeddings. **LayerNorm** (Basu et al., 2023) updates the trainable parameters of the LN blocks in each Transformer layer. **DiffFit** (Xie et al., 2023) updates both the bias terms and the LN blocks and inserts learnable factors to scale the features after the MSA and the MLP blocks. Instead of updating parameters, **SSF** (Lian et al., 2022) linearly adapts intermediate features, motivated by feature modulation (Huang & Belongie, 2017; Perez et al., 2018). For an intermediate feature $Z \in \mathbb{R}^{D \times (N+1)}$, SSF learns a $D$-dimensional scaling vector and a $D$-dimensional additive vector broadcasting to the tokens.

**Efficient selective parameter tuning.** Unlike the above category which directly updates parameters, this category learns *additive residuals* (*e.g.*, $\Delta W$) to the original parameters (*e.g.*, $W$). By injecting a low-rank constraint to the residuals, this category effectively reduces the learnable parameters. **LoRA** (Hu et al., 2021), arguably the most well-known approach, parameterizes the residuals by low-rank decomposition to update the Query/Value projection matrices $W_{Q/V} \in \mathbb{R}^{D \times D}$. Concretely, to update a $W \in \mathbb{R}^{D \times D}$ matrix, LoRA learns $W_{\text{down}} \in \mathbb{R}^{r \times D}$ and $W_{\text{up}} \in \mathbb{R}^{D \times r}$ with $r \ll D$, and forms the additive residual by $\Delta W = W_{\text{up}} W_{\text{down}} \in \mathbb{R}^{D \times D}$. **Factor Tuning (FacT)** (Jie & Deng, 2023) extends the idea of matrix decomposition into tensor decomposition. It stacks the $D \times D$ learnable matrices in all the Transformer layers into a 3D tensor and learns an additive residual parameterized by the well-established Tensor-Train (TT) (Oseledets, 2011) and Tucker (TK) (De Lathauwer et al., 2000) formulations.

More detailed descriptions of ViT and PETL methods can be found in Appendix B.

### 2.3 RELATED WORK AND COMPARISON

The community-wide enthusiasm for PETL has led to multiple survey articles (Yu et al., 2023a; Xin et al., 2024; Han et al., 2024). Meanwhile, several empirical, integrative, and theoretical studies were presented, mostly based on NLP tasks, attempting to provide a holistic understanding. (He et al., 2021a; Mao et al., 2021) provided unified views to methodologically connect PETL approaches. (Chen et al., 2022a; Ding et al., 2023; He et al., 2021b) and (He et al., 2022a) empirically compared PETL approaches on NLP and vision tasks, respectively, while (Fu et al., 2023) offered a theoretical stability and generalization analysis. Accuracy-wise, (Chen et al., 2022a; Ding et al., 2023; He et al., 2021b) found that PETL is robust to over-fitting and quite effective in NLP tasks under low-data regimes. *This is, however, not the case for vision tasks: (He et al., 2022a) showed that representative PETL approaches like LoRA and Adapter cannot consistently outperform either full fine-tuning or linear probing.* In terms of why PETL works, (Fu et al., 2023) framed PETL as sparse fine-tuning and showed that it imposes a regularization by controlling stability; (Ding et al., 2023; He et al., 2022a) framed PETL as (subspace) optimization; (Ding et al., 2023) further discussed the theoretical principle inspired by optimal control.

Our study strengthens and complements the above studies and offers new insights. First, we compared over ten PETL approaches, more than any of the above. We carefully tune the hyper-parameters, aiming to reveal the faithful accuracy of each approach. This is particularly important for the vision community because there have been no unifying references for PETL accuracy; simple approaches like BitFit have often been reported as quite inferior; the effectiveness of other approaches was reported quite discrepant from the study in NLP. Second, we go beyond a *competition* perspective to investigate a *complementary* perspective of PETL approaches. We show that different PETL approaches offer effective base learners for model ensembles. Third, we go beyond downstream accuracy to investigate PETL's effectiveness in maintaining out-of-distribution robustness. Fourth, we systematically analyze the results from low-shot and many-shot regimes and identify two distinct patterns among PETL, full fine-tuning, and linear probing, extending the understanding of PETL.

| Method | Natural | | | | | | | | Specialized | | | | | Structured | | | | | | | | | Overall Mean | Tunable Params |
|---|---|---|---|---|---|---|---|---|---|---|---|---|---|---|---|---|---|---|---|---|---|---|---|---|
| | CIFAR-100 | Caltech101 | DTD | Flowers102 | Pets | SVHN | Sun397 | Mean | Camelyon | EuroSAT | Resisc45 | Retinopathy | Mean | Clevr-Count | Clevr-Dist | DMLab | KITTI-Dist | dSpr-Loc | dSpr-Ori | sNORB-Azim | sNORB-Elev | Mean | | |
| Linear | 78.1 | 86.6 | 65.7 | 98.9 | 89.3 | 41.5 | 53.2 | 72.5 | 83.1 | 90.0 | 74.9 | 74.6 | 80.6 | 37.5 | 35.1 | 36.5 | 64.6 | 16.2 | 29.4 | 17.3 | 23.7 | 32.5 | 61.9 | 0 |
| Full | 62.4 | 89.9 | 61.9 | 97.4 | 85.8 | 88.9 | 36.8 | 76.7 | 81.6 | 88.1 | 81.6 | 73.6 | 81.2 | 56.2 | 60.9 | 48.2 | 77.9 | 68.5 | 46.6 | 31.0 | 28.3 | 52.2 | 70.0 | 85.8 |
| VPT-Shallow | 80.2 | 88.7 | 67.9 | 99.1 | 89.6 | 77.0 | 54.2 | 79.4 | 81.8 | 90.3 | 77.2 | 74.4 | 80.9 | 42.2 | 52.4 | 38 | 66.5 | 52.4 | 43.1 | 15.2 | 23.2 | 41.6 | 67.3 | 0.07 |
| VPT-Deep | 84.8 | 91.5 | 69.4 | 99.1 | 91.0 | 85.6 | 54.7 | 81.8 | 86.4 | 94.9 | 84.2 | 73.9 | 84.9 | 79.3 | 62.4 | 48.5 | 77.9 | 80.3 | 56.4 | 33.2 | 43.8 | 60.2 | 75.6 | 0.43 |
| BitFit | 86.5 | 90.5 | 70.3 | 98.9 | 91.0 | 91.2 | 54.2 | 82.6 | 86.7 | 95.0 | 85.3 | 75.5 | 85.6 | 77.2 | 63.2 | 51.2 | 79.2 | 78.6 | 53.9 | 30.1 | 34.7 | 58.5 | 75.6 | 0.1 |
| DiffFit | 86.3 | 90.2 | 71.2 | 99.2 | 91.7 | 91.2 | 56.1 | 83.2 | 85.8 | 94.1 | 80.9 | 75.2 | 84.0 | 80.1 | 63.4 | 50.9 | 81.0 | 77.8 | 52.8 | 30.7 | 35.5 | 59.0 | 75.4 | 0.14 |
| LayerNorm | 86.0 | 89.7 | 72.2 | 99.1 | 91.4 | 90.0 | 56.1 | 83.0 | 84.7 | 93.8 | 83.0 | 75.2 | 84.2 | 77.5 | 62.2 | 49.9 | 78.1 | 78.0 | 52.1 | 24.3 | 34.4 | 57.1 | 74.7 | 0.04 |
| SSF | 86.6 | 89.8 | 68.8 | 99.1 | 91.4 | 91.2 | 56.5 | 82.8 | 86.1 | 94.5 | 83.2 | 74.8 | 84.7 | 80.1 | 63.6 | 53.0 | 81.4 | 85.6 | 52.1 | 31.9 | 37.2 | 60.6 | 76.0 | 0.21 |
| Pfeif. Adapter | 86.3 | 91.5 | 72.1 | 99.2 | 91.4 | 88.5 | 55.7 | 83.0 | 86.2 | 95.5 | 85.3 | 76.2 | 85.8 | 83.1 | 65.2 | 51.4 | 80.2 | 83.3 | 56.6 | 33.8 | 41.1 | 61.8 | 76.9 | 0.67 |
| Houl. Adapter | 84.3 | 92.1 | 72.3 | 98 | 91.7 | 90.0 | 55.4 | 83.2 | 88.7 | 95.3 | 86.5 | 75.2 | 86.4 | 82.9 | 63.6 | 53.8 | 79.6 | 84.4 | 54.3 | 34.2 | 44.3 | 62.1 | 77.2 | 0.77 |
| AdaptFormer | 85.8 | 91.8 | 70.5 | 99.2 | 91.8 | 89.4 | 56.7 | 83.2 | 86.8 | 95.0 | 86.0 | 76.3 | 86.2 | 82.9 | 64.1 | 52.8 | 80.0 | 84.7 | 53.0 | 33.0 | 41.4 | 61.5 | 76.9 | 0.46 |
| RepAdapter | 86.0 | 92.5 | 69.1 | 99.1 | 90.9 | 90.9 | 55.4 | 82.9 | 86.9 | 95.3 | 86.0 | 75.4 | 85.9 | 82.5 | 63.5 | 51.4 | 80.2 | 85.4 | 52.1 | 35.7 | 41.7 | 61.6 | 76.8 | 0.53 |
| Convpass | 85.0 | 92.1 | 72.0 | 99.3 | 91.3 | 90.8 | 55.9 | 83.5 | 87.7 | 95.8 | 85.9 | 75.9 | 86.3 | 82.3 | 65.2 | 53.8 | 78.1 | 86.5 | 55.3 | 38.6 | 45.1 | 63.1 | 77.6 | 0.49 |
| LoRA | 85.7 | 92.6 | 69.8 | 99.1 | 90.5 | 88.5 | 55.5 | 82.6 | 87.5 | 94.9 | 85.9 | 75.7 | 86.0 | 82.9 | 63.9 | 51.8 | 79.9 | 86.6 | 47.2 | 33.4 | 42.5 | 61.0 | 76.5 | 0.55 |
| FacT_TT | 85.8 | 91.8 | 71.5 | 99.3 | 91.1 | 90.8 | 55.9 | 83.4 | 87.7 | 94.9 | 85.0 | 75.6 | 85.8 | 83.0 | 64.0 | 49.0 | 79.3 | 85.8 | 53.1 | 32.8 | 43.7 | 61.3 | 76.8 | 0.13 |
| FacT_TK | 86.2 | 92.5 | 71.8 | 99.1 | 90.1 | 91.2 | 56.2 | 83.4 | 85.8 | 95.5 | 86.0 | 75.7 | 85.8 | 82.7 | 65.1 | 51.5 | 78.9 | 86.7 | 53.1 | 40.8 | 40.6 | 60.8 | 76.6 | 0.23 |
| Relative Std Dev | 0.81 | 1.13 | 1.78 | 0.34 | 0.54 | 1.82 | 1.24 | 0.54 | 1.20 | 0.59 | 1.95 | 0.83 | 0.94 | 2.67 | 1.50 | 3.22 | 1.37 | 4.11 | 4.46 | 11.02 | 9.30 | 2.70 | 1.09 | - |

Table 1: Results on VTAB-1K (19 tasks from 3 groups). Based on the accuracy among PETL, linear probing, and full fine-tuning, we find two task groups (purple and orange), as discussed in section 6.

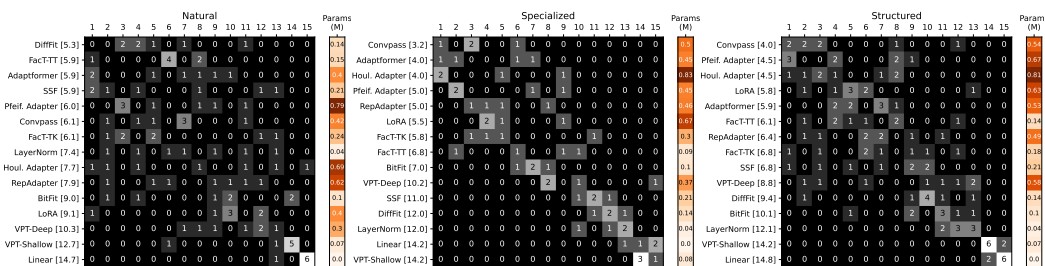

Figure 2: Ranking frequency of 15 methods (14 PETL + linear probing) for three groups in VTAB-1K. Element $(i, j)$ is the number of times method $i$ ranks $j^{th}$ in each group. Methods are ordered by mean ranks (in brackets). The parameters column shows the # of trainable parameters in millions. More details are in Appendix C.

# 3 HOW DO PETL METHODS PERFORM IN LOW-SHOTS REGIME?

Pre-trained models are meant to ease downstream applications. One representative scenario is low-shot learning: supervised fine-tuning with a small number of examples per class. Indeed, low-shot learning has been widely used to evaluate PETL performance.

**Dataset. VTAB-1K** (Zhai et al., 2019) consists of 19 image classification tasks from three groups. The **Natural** group comprises natural images captured with standard cameras. The **Specialized** group contains images captured by specialist equipment for remote sensing and medical purposes. The **Structured** group evaluates the scene structure comprehension, such as object counting and 3D depth estimation. Following Zhai et al. (2019), we perform an 80/20 split on the **1000** training images in each task for hyperparameter tuning. The reported top1 accuracy is obtained by training on the 1000 training images and evaluating on the original test set.

**Methods.** We consider linear probing, full fine-tuning, and **14** PETL methods including **2** prompt-based (Jia et al., 2022), **5** adapter-based (Houlsby et al., 2019; Pfeiffer et al., 2021; Chen et al., 2022b; Jie & Deng, 2022; Luo et al., 2023), **4** Direct selective (Zaken et al., 2022; Basu et al., 2023; Xie et al., 2023; Lian et al., 2022), and **3** Efficient selective (Hu et al., 2021; Jie & Deng, 2023). Please refer to subsection 2.2 for details.

**Setup.** We employ the ViT-B/16 model (Dosovitskiy et al., 2020) pre-trained on ImageNet-21K (Deng et al., 2009) as the backbone. The prediction head is randomly initialized for each dataset. Images are resized to $224 \times 224$. We systematically tune 1) learning rate, 2) weight decay, and 3) approach-specifics like the size of PETL parameters which are often left intact. **We set a cap for 3), $\leq 1.5\%$ of ViT-B/16.** We also turn the drop path rate (Huang et al., 2016) on (*e.g.*, 0.1) or off (*i.e.*, 0). A detailed hyperparameter search grid and additional training details are provided in Appendix A.1.

**Results.** As shown in Figure 1a and Table 1, PETL methods generally outperform both linear probing and full fine-tuning across datasets. Additionally, under fair hyper-parameter tuning, we surprisingly found that most PETL methods perform similarly as the relative standard deviations (divided by

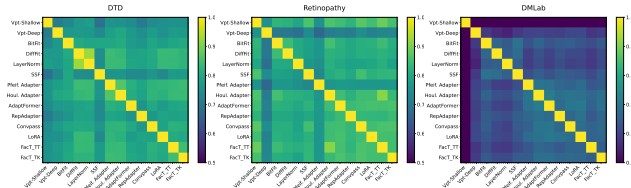

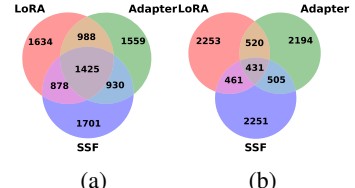

(a)  (b)

Figure 3: Prediction similarity analysis: element $(i, j)$ shows the percentage of samples that method $i$ and method $j$ have the same predictions. Although different methods achieve similar accuracy, they have diverse predictions. Details in Appendix C.

Figure 4: LoRA, Adapter and SSF fine-tuned on CIFAR100(VTAB-1K). (a): correct prediction overlap for the 5K most confident data. (b): wrong prediction overlap for the 5K least confident data. Details in Appendix C.

the means) in all three groups are quite low. Simple methods (*e.g.*, Bitfit) and PETL methods originally proposed for NLP (*e.g.*, LoRA and Adapter), which were previously reported as inferior due to un-optimized implementations and hyperparameter tuning, now demonstrate competitive performance with SOTA visual PETL methods. To understand the relative advantages of different approaches, we provide the ranking frequency of PETL methods across different groups in Figure 2, where the element $(i, j)$ in each ranking matrix represents the frequency that method $i$ ranks $j^{th}$ in each group. Methods are ordered by their mean ranks (in brackets), and the parameters column indicates the number of trainable parameters in millions. In the natural group, simpler methods with fewer trainable parameters—such as DiffFit and Fact-TT—offer a cost-effective solution without compromising performance. Conversely, in the specialized and structured groups, methods with more parameters generally yield better performance. We hypothesize that this performance discrepancy arises from the **domain affinity** between the pre-trained domain (ImageNet) and the downstream domains. The natural group, sharing a stronger affinity with ImageNet, allows simpler methods like BitFit to adjust the features effectively. In contrast, the specialized and structured groups necessitate more complex methods with more trainable parameters to bridge the domain gap.

**Recipes.** In low-shot regimes, when the downstream data are similar to the pre-trained data, simple methods (*e.g.*, DiffFit) with decent accuracy and much fewer parameters are preferred—aligning with Occam's razor. When there is a substantial domain gap, complex methods with higher accuracy (but more parameters) become competitive. Learning with low-shot data is prone to over-fitting. We find that if the drop path rate — which stochastically drops a transformer block per sample (Huang et al., 2016) — is set not as default (*i.e.*, nonzero), all the methods can benefit from such a regularization, as shown in Figure 10 in the Appendix.

## 4 PETL Approaches Offer Complementary Information

The previous section demonstrates that all PETL methods perform similarly across various domains. Given that different PETL methods are trained on the same downstream data using the same backbone and achieve comparable accuracy, one might expect them to learn similar knowledge from the data, resulting in similar predictions. Contrary to this expectation, our findings below reveal that different PETL methods acquire **distinct** and **complementary** knowledge from the same downstream data, even when built upon the same backbone, leading to diverse predictions.

We start by analyzing their prediction similarity on the same dataset in VTAB-1K. It is expected that their predictions are similar for datasets with very high accuracy, such as Flowers102 (avg 99.1%) and Caltech101 (avg 91.4%). Beyond them, we find that most PETL methods show diverse predictions in other datasets in VTAB-1K. Figure 3 shows the prediction similarities between 14 PETL methods in DTD, Retinopathy, and DMLab, which belong to natural, specialized, and structured groups, respectively. In DTD and Retinopathy, most methods differ in about 20% of their predictions, while in DMLab, this difference increases to approximately 35%, even though they achieve similar accuracies. This prediction diversity may be attributed to the different *inductive biases* (Neyshabur et al., 2014) of PETL methods — they explicitly select specific parameters to update or insert different modules at various locations within the model. More analyses and details about Figure 3 are offered in Appendix C.

Such diverse predictions across methods open up the possibility of leveraging their heterogeneity for further improvement. The most straightforward approach is ensemble (Gontijo-Lopes et al., 2021), *e.g.*, average logits over methods. Figure 5 demonstrates the ensemble performance gain over all the PETL methods in each dataset, where we use the worst PETL method as the baseline. Due to the diverse predictions across methods, the ensemble can provide consistent gain.

Also, we analyze if PETL methods make similar correct predictions for high-confidence samples and similar mistakes for low-confidence samples. Figure 4 shows the correct prediction overlap for the 5K most confident samples (per method) and the wrong prediction overlap for the 5K least confident samples (per method). For demonstration purposes, we select one method from each PETL category (LoRA, Adapter, SSF) and they are fine-tuned on CIFAR-100 in VTAB-1K. Methods within the same category also show diverse predictions (Appendix C). Since they make different predictions in both high and low-confidence regimes, this paves the way for new possibilities of using different PETL methods to generate **diverse pseudo-labels** for semi-supervised learning (Yang et al., 2022), domain adaptation (Farahani et al., 2021), and transfer learning (Zhuang et al., 2020).

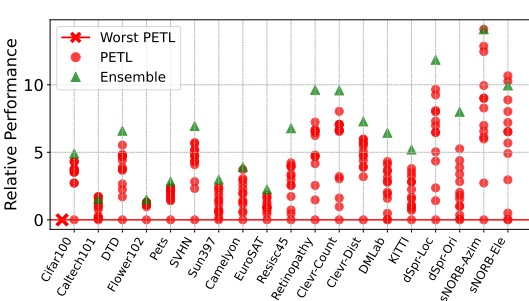

Figure 5: Ensemble (avg logits) provides consistent gain on most datasets thanks to the diverse prediction. Details in Appendix C.

## 5   HOW DO PETL METHODS PERFORM IN MANY-SHOT REGIME?

Recent works in NLP Chen et al. (2022a) have indicated that PETL methods may not perform as competitively as full fine-tuning when data is abundant. We thus aim to investigate PETL's performance in many-shot regimes by addressing the following questions: (1) Should we use PETL or full fine-tuning when data is sufficient? (2) How should we adjust the number of trainable parameters for PETL methods in many-shot regimes?

**Dataset.** We select one representative dataset from each of the natural, specialized, and structured groups in VTAB: (1) CIFAR-100 Krizhevsky et al. (2009), a natural image dataset comprising 50K training images across 100 classes; (2) RESISC Cheng et al. (2017), a remote sensing dataset for scene classification with 25.2K training samples across 45 classes; and (3) Clevr-Distance Zhai et al. (2019); Johnson et al. (2017), a synthetic image dataset for predicting the depth of the closest object from the camera with 6 depth classes and 70K samples. The reported results are obtained by training on the **full** training set and evaluating on the original test set.

**Setup.** The model setup follows the VTAB-1K experiment. More details about setup and hyperparameter search are provided in Appendix A.

**Results.** In many-shot regimes, with sufficient downstream data, full fine-tuning may catch up and eventually outperform PETL methods. However, from Figure 6, we found that even in many-shot regimes, PETL can achieve **comparable results with full fine-tuning**, even just using $2\%$ of fine-tuning parameters. (The performance gain, however, quickly **diminishes and plateaus after 5%** of tunable parameters.) By comparing the results on the domain-close CIFAR-100 and domain-different RESISC and Clevr, we have some further observations. On the one hand, downstream tasks with larger domain gaps suggest the need to update, perhaps many, parameters to obtain high accuracy. With sufficient downstream data, full fine-tuning is less prone to over-fitting and indeed attains a high accuracy. But interestingly, PETL methods, with only $2 \sim 5\%$ of tunable parameters, achieve similar accuracy, suggesting that its design principle does offer sufficient effective capacity for the model to learn Zhang et al. (2021). On the other hand, downstream tasks with smaller domain gaps suggest that the pre-trained model had learned sufficient knowledge about them; fully fine-tuning it thus risks washing such knowledge away. In fact, we found that PETL notably outperforms full fine-tuning on CIFAR-100, suggesting it as a more robust *transfer learning* algorithm for downstream tasks.

**Recipes.** In many-shot regimes, PETL methods with sufficient parameters ($2 \sim 5\%$) appear more favorable than full fine-tuning and linear probing. On the one hand, they achieve comparable and even

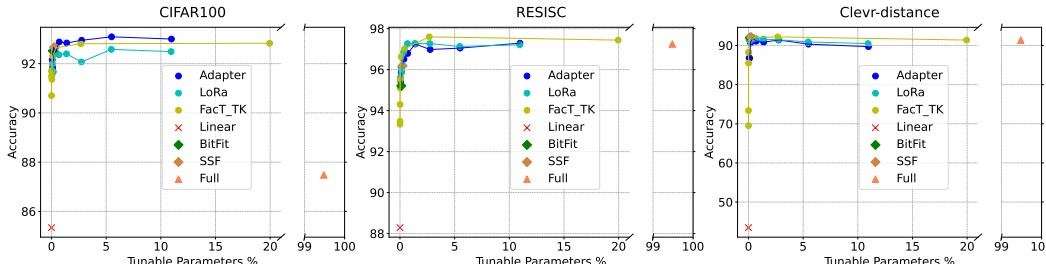

Figure 6: PETL accuracy in many-shot regimes, with different parameter sizes (X-axis) on three datasets from different domains. Even 2%-5% trainable parameters allow the models to have sufficient capacity to learn from full data. (Details are in Appendix C)

better accuracy than full fine-tuning. On the other hand, the tunable parameters remain manageable. The parameter efficiency of PETL also often implies less training GPU memory usage and training time, making PETL methods a favorable alternative in many-shot regimes. For a downstream domain that is close to the pre-training domain, PETL shows much pronounced *transferability*. For a downstream domain that is quite different, the limited tunable parameters (controversially, $2 \sim 5\%$ already amounts to a few million) already allow the model to learn sufficiently.

# 6 WHY DO PETL METHODS WORK? [1]

Putting together section 3 and section 5, we identify two distinct patterns regarding the performance among linear probing, full fine-tuning, and PETL. Within 19 VTAB-1K tasks, we see: (1) Full fine-tuning outperforms linear probing. As linear probing reflects the pre-trained feature quality for downstream tasks, case (1) suggests the necessity to update the backbone to close the gap between pre-trained and downstream domains. (2) Linear probing surpasses full fine-tuning, suggesting the pre-trained features are good enough (at least in a low-shot scenario). Recklessly updating them may risk over-fitting. Figure 7 (a-b) summarizes the low-shot accuracy comparison based on the categorization above; each line corresponds to one task. Linear probing, PETL, and fine-tuning are located in order, from left to right, to reflect their tunable parameter sizes. PETL's superiority in both cases showcases its **capacity** to learn and its **regularization role** to prevent over-fitting.

We also draw the many-shot accuracy in Figure 7 (c-d) based on the same categorization: RESISC and Clevr in case (1) , and CIFAR-100 in case (2) . In the many-shot setting, full fine-tuning consistently outperforms linear probing, which seems to suggest *no more risk of over-fitting*. However, on CIFAR-100 (Figure 7 (d)), we again see a noticeable gap between PETL and full fine-tuning, just like in Figure 7 (b). Such a concave shape reminds us of the long-standing under-fitting-over-fitting curve, suggesting that even with sufficient downstream data, full fine-tuning still risks over-fitting.

Taking into account PETL's comparable performance to full fine-tuning on RESISC and Clevr with large domain gaps, we conclude — PETL succeeds as a **high-capacity** learner equipped with an **effective regularizer**; the two roles trade-offs well such that PETL can excel in both low-affinity and high-affinity domains under both low-shot and many-shot settings.

# 7 ARE PETL METHODS MORE ROBUST TO DISTRIBUTION SHIFTS?

Large pre-trained models such as CLIP Radford et al. (2021) and ALIGN Jia et al. (2021) have demonstrated unprecedented accuracy across a range of data distributions when performing zero-shot inference. However, recent studies Wortsman et al. (2022); Radford et al. (2021) have shown that fine-tuning on downstream data, while significantly boosting performance on the target distribution, often compromises the model's robustness to distribution shifts. Given that PETL only updates a

---

[1]Our intention is not to offer a definitive conclusion about why PETL works. As discussed in subsection 2.3, there is currently no universally agreed-upon explanation for the effectiveness of PETL. We hope our empirical findings will contribute to the ongoing efforts to understand the underlying principles of PETL methods.

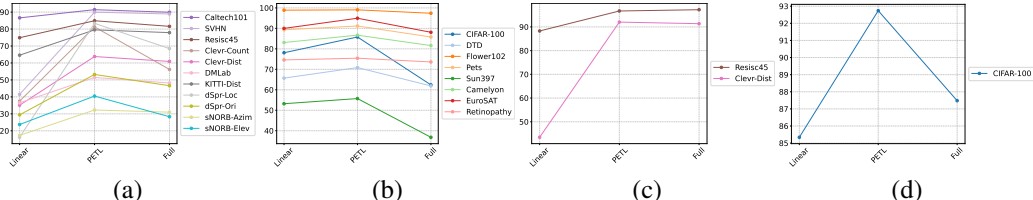

Figure 7: (a): VTAB-1K tasks in case (1) , PETL > full > linear . (b) VTAB-1K tasks in case (2), PETL > linear > full. (c) RESISC & Clevr in case (1) with enough data, PETL ≈ full > linear. (d) CIFAR in case (2) with enough data, PETL > full > linear. Within each figure, left for linear, middle for PETL, and right for full. More details are in Appendix C.

| | Full | BitFit | Layer-Norm | Houl. Adapter | Adapt-Former | Rep-Adapter | Convpass | LoRA | FacT_TK |
|---|---|---|---|---|---|---|---|---|---|
| 100-shot ImageNet | 75.0 | 75.27 | 74.8 | 75.0 | 75.6 | 76.5 | 76.3 | 76.6 | 74.7 |
| Avg. distribution shift Acc | 42.5 | 55.4 (12.9)↑ | 55.9 (13.4)↑ | 56.9 (14.4)↑ | 56.1 (13.6)↑ | 56.2 (13.7)↑ | 54.7 (12.2)↑ | 55.9 (13.4)↑ | 56.1 (13.6)↑ |

Table 2: The "Avg. distribution shift Acc" denotes the average performance of ImageNet-(V2, S, R, A) evaluated on the CLIP model fine-tuned on ImageNet. (↑) indicates the gain over full fine-tuning.

limited number of parameters in the model, we investigate whether PETL can offer a more robust alternative to full fine-tuning for pre-trained models.

**Dataset.** We use 100-shot ImageNet-1K as our target distribution, with each class containing 100 images. Following Wortsman et al. (2022), we consider 4 natural distribution shifts from ImageNet: **ImageNet-V2** Recht et al. (2019), a new ImageNet test set collected with the original labeling protocol; **ImageNet-R** Hendrycks et al. (2021a), renditions for 200 ImageNet classes; **ImageNet-S** Gao et al. (2023), sketch images for 1K ImageNet classes; **ImageNet-A** Hendrycks et al. (2021b), a test set of natural images misclassified by a ResNet-50 He et al. (2015) for 200 ImageNet classes.

**Setup.** We focus on the CLIP ViT-B/16 model, which comprises a visual encoder and a text encoder, pre-trained via contrastive learning on image-text pairs. Following Wortsman et al. (2022), we add an FC layer as the prediction head with zero-initialized bias and initialize weights using the class label text embedded by the text encoder. Subsequently, we discard the text encoder and apply PETL methods to the visual encoder, fine-tuning only the PETL modules and the head. More details about the CLIP model and experiment setup can be found in Appendix A.1.

**Results.** As shown in Table 2, while some PETL methods may not surpass full fine-tuning on the target distribution, they consistently demonstrate more robust performance on distribution shift data. This is likely because PETL updates only a small fraction of the parameters, thus preserving the robust features of the foundation models. Given the similar target distribution performance, should we blindly use PETL methods for more robustness?

**Weight-space ensembles (WiSE) for PETL.** WiSE (Wortsman et al., 2022), which linearly interpolates the full fine-tuned and original models, is a popular fine-tuning approach to enhance robustness. We explore whether WiSE can enhance the robustness of PETL. To apply WiSE to PETL, we first linearly interpolate the prediction head with a mixing coefficient $\alpha$. For direct selective tuning methods (*e.g.*BitFit), we directly interpolate with the original model. Since most Adapter-based methods have residual connections, we can multiply the adapter modules with $\alpha$ to control their strengths. A similar approach can be applied to efficient selective methods (*e.g.*LoRA) as they learn additive residuals to the original parameters. As shown in Figure 1b (more results in Appendix C), WiSE improves both fine-tuning and distribution shift performance of PETL methods. Interestingly, even though full fine-tuning is generally less robust than PETL methods, applying WiSE allows it to achieve better performance in both target distribution and distribution shift data, which suggests a promising research direction for robust PETL.

## 8 CONCLUSION

We conduct a unifying empirical study of parameter-efficient fine-tuning (PETL), an emerging topic in the large model era. We have several new insights and implications, including PETL methods' complementary expertise, suitable application regimes, and robustness to domain shifts. We expect our study to open new research directions and serve as a valuable user guide in practice.

## 9 REPRODUCIBILITY STATEMENT

We have made extensive efforts to ensure the reproducibility of our results. All 14 PETL algorithms and two baseline models, along with data preprocessing routines and data loaders for all datasets—including 19 low-shot datasets, 3 many-shot (full) datasets, and 5 robustness-related datasets—are implemented within a systematic and extensible framework. This framework is designed to facilitate the easy addition of new PETL methods and datasets, modification of backbones, and incorporation of additional scenarios, serving as a convenient tool for future research. Detailed explanations of our implementations, raw results for all experiments, and commands to reproduce the results are thoroughly documented in the README file. We provide the anonymous source code in the supplementary material.

## 10 ETHICS STATEMENT

Our study provides a unifying study of PETL in visual recognition. We expect it to serve as a valuable practical user guide to benefit society. Specifically, fine-tuning large models needs significant computation. A unifying study of PETL will ease end-users to apply more parameter-efficient and computation-efficient ways for fine-tuning. To our knowledge, our paper does not introduce any additional negative societal impacts compared to existing papers on PETL.

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
