# OpenReview forum: "Lessons Learned from a Unifying Empirical Study of Parameter-Efficient Transfer Learning (PETL) in Visual Recognition"
_ICLR.cc/2025/Conference — ICLR 2025 Conference Withdrawn Submission_

### Official Review · Reviewer_dffk · 2024-10-23

**Soundness:** 2
**Presentation:** 3
**Contribution:** 2
**Rating:** 3
**Confidence:** 4

**Summary:**

This paper aims to systematically investigate the performance of parameter-efficient transfer learning (PETL) methods in visual recognition tasks. By tuning hyperparameters of representative PETL methods, the authors compare their performance on the low-shot task VTAB-1K and other datasets. It was found that while different PETL methods exhibit similar accuracy, they differ in terms of high-confidence predictions and errors, which presents an opportunity for the application of ensemble methods. Additionally, the study shows that PETL performs well in multi-sample tasks and outperforms full model fine-tuning under distribution shift conditions.

**Strengths:**

The authors conducted extensive experiments, systematically adjusting hyperparameters to fairly compare the performance of various PETL methods, particularly in low-shot tasks.

**Weaknesses:**

1 While the authors conducted numerous experiments, there is a lack of in-depth theoretical analysis of the underlying mechanisms of PETL methods. For instance, in lines 132–133, the authors mention, "In sum, PETL succeeds as a high-capacity learner equipped with an effective regularizer," yet they seem to lack theoretical justification.

2 The experiments are primarily based on the VTAB-1K dataset. Although it covers 19 tasks, it lacks validation on more challenging or widely-used datasets.

3 Although the authors state in lines 139–140 that "our study does not consider computation-specific properties like memory usage and FLOPS," these are critical factors when evaluating the practical application of the methods, especially in large-scale scenarios.

4 The results show that all PETL methods perform similarly after proper hyperparameter tuning. However, it is unclear whether this similarity is due to task limitations or inherent limitations of PETL methods, and the paper lacks further discussion on this point.

5 The experimental methods selected by the authors are relatively outdated, for example, SSF from NeurIPS 2022 and FACT from AAAI 2023.

**Questions:**

See Weaknesses.

---

### Official Review · Reviewer_cuwg · 2024-10-29

**Soundness:** 3
**Presentation:** 4
**Contribution:** 3
**Rating:** 5
**Confidence:** 5

**Summary:**

This article experimentally analyses the performance of visual PEFT methods on image classification tasks, encompassing a variety of PEFT methods. The paper also raises several issues of concern in the field of PEFT and presents visionary ideas.

**Strengths:**

1. The writing of the article is quite good and the article is logical.
2. The experiments on the classification task seem to be  adequate.
3. The charts and graphs are more beautiful and properly presented.
4. Supplementary materials are substantial.

**Weaknesses:**

1. Title: The authors use "Visual Recognition" in the title. In CVPR's area options, this area includes multiple tasks such as classification, detection, segmentation, etc., but it seems that the paper is only validated on the classification task. I suggest adding other tasks to the paper, as has been done in several recent PEFT works [1-3].
[1] 1% vs 100%: Parameter-efficient low rank adapter for dense predictions.
[2] Pro-tuning: Unified prompt tuning for vision tasks.
[3] Adapter is all you need for tuning visual tasks.
[4] Parameter-efficient is not sufficient: Exploring parameter, memory, and time efficient adapter tuning for dense predictions.

2. Section 2.1 could be more concise in its introduction to ViT.
3. Related work: the authors perhaps left out some of the most recent work of PEFT.
4. Experiments: Most of the methods in baseline outperform FULL on VTAB-1K. Since this is an analysis paper, experiments on more challenging tasks (e.g., COCO, ADE20K, etc.) would be more convincing.

I would like to increase the score if the authors could solve the problems above properly.

**Questions:**

See pros and cons.

---

### Official Review · Reviewer_NiRP · 2024-11-03

**Soundness:** 2
**Presentation:** 3
**Contribution:** 2
**Rating:** 6
**Confidence:** 4

**Summary:**

This paper conducts an exhaustive study of PETL methods based on ViT in the field of vision recognition. Through experiments, the paper analyzes the applicable scenarios and compares the advantages and disadvantages of various PETL methods. Specifically, the authors argue that after sufficient parameter tuning, the performance of various PETL methods in low-shot scenarios becomes relatively similar, with fewer trainable parameters performing better in similar domains and the opposite in more dissimilar domains. However, the prediction results vary, so better outcomes can be achieved through ensemble methods. When data is abundant, PETL methods perform better in similar domains but slightly worse than full fine-tuning methods in more dissimilar domains. The authors also analyze the robustness of PETL methods and conclude that PETL methods are more robust than full fine-tuning methods. Additionally, the authors point out that combining PETL methods with WiSE methods leads to opposite conclusions, suggesting that this is a worthwhile research direction for PETL methods.

**Strengths:**

* The writing of this paper is good, making it easy for readers to understand.
* The experiments in this article are comprehensive, providing a detailed comparison of the performance of PETL methods, analyzing their applicable scenarios, and drawing appropriate conclusions.
* This article is valuable to the community, as readers can gain a thorough understanding of the PETL direction from this paper.

**Weaknesses:**

* The conclusions presented in the article, such as the advantages of PETL methods in low-shot scenarios and on data from closer domains, seem to be known knowledge within the field. For example, some methods attempt to reduce the distance between domains to improve the performance of domain transfer methods [1].
* This paper focuses more on summarizing existing methods and does not analyze the future development trends of PETR methods. Although the authors briefly discuss some interesting phenomena when combining with WiSE methods.

> [1] FDA: Fourier Domain Adaption for Semantic Segmentation. CVPR 2020.

**Questions:**

* The authors point out that the results of different PETR methods are complementary. Do the prediction results of different types of methods have their own trends, for example, whether adapter-based methods tend to make similar incorrect predictions?
* If the authors could discuss the application of PETR methods in some of the latest directions, such as image generation, it would enrich the content of the article.

---

### Official Review · Reviewer_bL5F · 2024-11-03

**Soundness:** 2
**Presentation:** 3
**Contribution:** 2
**Rating:** 5
**Confidence:** 4

**Summary:**

This paper presents a systematic empirical study of parameter-efficient transfer learning (PETL) methods in the context of vision transformers (ViT). The authors investigate various PETL approaches, analyze their performance in low-shot and many-shot scenarios, and examine their robustness to distribution shifts. Key contributions include a comparative analysis of different PETL methods, insights into their complementary nature, and guidelines for practitioners regarding their effective application. The study aims to clarify when and how to use PETL methods effectively across different domains.

**Strengths:**

+The paper addresses important questions in the field of PETL, contributing meaningfully to the ongoing discourse.

+The finding that weight-space ensembles (WiSE) could enhance performance on the target distribution is intriguing and merits further investigation.

**Weaknesses:**

While this paper explores several important questions and presents key findings, some analyses appear somewhat subjective and lack the detailed experimentation necessary to be fully convincing. Additionally, certain discussions align closely with existing knowledge and do not provide substantial new insights.

-The analysis of PETL performance in many-shot regimes closely resembles studies involving SSF [1], Bi-Adapter [2], and supplementary materials of GPS [3], which may limit the originality of the contributions.

-The empirical recommendations for practitioners regarding when and how to use PETL methods could be more actionable. For instance, the concept of domain affinity is not clearly defined—could it be quantified in a statistical manner?

-The paper raises questions about how Occam’s razor applies to similar downstream data while seemingly failing in scenarios with significant domain gaps. A more thorough exploration of this aspect would enhance understanding.

-Some discussions, particularly around the robustness of PETL methods to distribution shifts, lack depth and do not offer meaningful new conclusions or clear differentiation from existing literature, as noted in lines 519-520.

-On the other hand, interesting findings such as the performance improvements from using weight-space ensembles (WiSE) in target distributions compared to direct fine-tuning are worth further investigation.

-The recommendations are largely based on experiments with classification tasks, which may limit their generalizability across a broader range of applications, potentially diminishing the overall significance of the contributions.

[1]Scaling & Shifting Your Features: A New Baseline for Efficient Model Tuning
[2]Revisiting the Parameter Efficiency of Adapters from the Perspective of Precision Redundancy
[3]Gradient-based Parameter Selection for Efficient Fine-Tuning

**Questions:**

Please refer to the weaknesses. In addition, I have the following two questions:

1) In line 366, the paper states that "PETL methods acquire distinct and complementary knowledge." Could this distinct and complementary knowledge arise from different random initializations of the tunable parameters? Investigating how various initializations influence the acquisition of this distinct and complementary knowledge could be beneficial.

2) When dealing with a new dataset or task, how should practitioners choose among the different PETL methods based on the analysis and recipes provided in the paper?

---

### Note · Authors · 2024-11-15

I have read and agree with the venue's withdrawal policy on behalf of myself and my co-authors.